# Radiation Damage of Reactor Pressure Vessel Steels Studied by Positron Annihilation Spectroscopy—A Review

**Vladimír Slugeň [1],\*, Stanislav Sojak [1], Werner Egger [2], Vladimir Krsjak [1]**, **Jana Simeg Veternikova [1] and Martin Petriska [1]**

[1] Institute of Nuclear and Physical Engineering, Slovak University of Technology, Ilkovičova 3, 81219 Bratislava, Slovakia; stanislav.sojak@stuba.sk (S.S.); vladimir.krsjak@stuba.sk (V.K.); jana.veternikova@stuba.sk (J.S.V.); martin.petriska@stuba.sk (M.P.)

[2] Faculty Applied Physics and Measurement, University of Bundeswehr, Werner Heisenberg Weg 39, D-85577 Munich, Germany; Werner.Egger@unibw.de

\* Correspondence: vladimir.slugen@stuba.sk; Tel.: +421-915-837843

**Abstract:** Safe and long term operation of nuclear reactors is one of the most discussed challenges in nuclear power engineering. The radiation degradation of nuclear design materials limits the operational lifetime of all nuclear installations or at least decreases its safety margin. This paper is a review of experimental PALS/PLEPS studies of different nuclear reactor pressure vessel (RPV) steels investigated over last twenty years in our laboratories. Positron annihilation lifetime spectroscopy (PALS) via its characteristics (lifetimes of positrons and their intensities) provides useful information about type and density of radiation induced defects. The new results obtained on neutron-irradiated and hydrogen ions implanted German steels were compared to those from the previous studies with the aim to evaluate different processes (neutron flux/fluence, thermal treatment or content of selected alloying elements) to the microstructural changes of neutron irradiated RPV steel specimens. The possibility of substitution of neutron treatment (connected to new defects creation) via hydrogen ions implantation was analyzed as well. The same materials exposed to comparable displacement damage (dpa) introduced by neutrons and accelerated hydrogen ions shown that in the results interpretation the effect of hydrogen as a vacancy-stabilizing gas must be considered, too. This approach could contribute to future studies of nuclear fission/fusion design steels treated by high levels of neutron irradiation.

**Keywords:** radiation damage; reactor pressure vessel steels; neutron embrittlement; hydrogen ion implantation; positron annihilation

## 1. Introduction

The reactor pressure vessels (RPV) are the key and most critical components in most nuclear power plants (NPP). These components are considered irreplaceable. Several world-wide PLIM/PLEX (Plant Lifetime Management and Extension) meetings were focused on this issue. If mechanical properties of RPV steels degrade below a certain limit, they can be considered as the life-limiting factor of the safe operation of the NPP unit. So called 'design basis life', planned for 30–40 years of operation, was not based on experimental analyses of construction material degradation, but in many cases based on fatigue usage calculations. Newest considerations of NPPs life-time prolongation are supported by long-term operational experiences, including the surveillance programs evaluation as well as results of many material analyses performed on specially treated specimens irradiated in research reactors. As a result of these studies the NPP lifetime increases and newly designed plants could be in operation for

60–80 years [1]. The current target for most plants in many countries in Europe, Japan and the USA (with re-licensing) is lifetime extension up to 60 or more years. Degradation process of RPV steels implies to safety concerns limiting operating time. Based on [1,2] the most important degradation factors are: (i) thermal ageing, (ii) fast neutron embrittlement, (iii) thermal embrittlement, (iv) fatigue and (v) corrosion.

Structural integrity of RPVs should be guaranteed throughout the whole operating lifetime for all normal operation and accident conditions, as well as for non-designed transients such as pressurized thermal shock (PTS). Neutron irradiation degrades the mechanical properties of RPV steels, and the extent of the degradation is determined by the type and structure of the steel, and other factors such as neutron fluence, irradiation temperature, neutron flux and chemical composition [1]. The most sensitive location in the RPV is the region adjacent to the reactor core (termed the beltline region). Welds and their heat affected zones (HAZs) in this region are particularly important since these regions have a higher probability of having flaws [2].

The selection of appropriate steels for RPV depends mainly on the design of the reactor (coolant type, operational temperatures, neutron flux/fluence, etc.). In pressurized water reactors (PWR), there is necessary to have regard for the chemical composition of the materials which must be resistant to corrosive influences of the coolant–water with boric acid at high temperature and pressure. The neutron flux at the wall of the RPV refers to a necessary level of purity in the chemical composition of the material due to transmutation effect and following degradation of this material. The basic required properties for the RPV materials are the following:

Mechanical properties–resistance and strength at higher temperatures; creep strength at higher temperatures; resistance to thermal stresses, cyclic stress and brittle fracture;

Corrosion resistance–resistance to degradation of the material surface due to corrosive materials (such as $H_2O$, $O_2$, $H_3BO_3$);

Thermal properties–affect heat transfer and thermal stresses including small thermal dilatations;

Nuclear-physical properties–low cross sections for neutrons, radiation stability;

High purity material–reducing the induced radioactivity (an advantage when decommissioning and dismantling will be actual);

Availability and price;

Technological properties–acceptable mechanical treatment and jointing.

The dominant element in RPV steel is iron. The crystal structure of the iron (arrangement of iron crystals of the crystal lattice) is BCC (body-centered cubic) up to the temperature of 906 °C. There is a change to the FCC (face-centered cubic) crystal structure at higher temperatures in the range of 906–1403 °C and from the temperature of 1403 °C is BCC structure formed again. In the crystal materials lattice defects reflect an irregularity of the crystal lattice and also the crystal quality of the material. Light transmutation elements such as helium or hydrogen can easily diffuse to these crystal defects and they can influence the mechanical properties [2,3].

Challenges, which are up-to date for the newest reactor design materials, can be studied with various techniques. These include (but are not limited) in situ and ex situ TEM, ion beam/scanning electron microscopy (FIB/SEM), micro X-ray diffraction (XRD), atom probe tomography (APT), positron annihilation spectroscopy (PAS), synchrotron techniques, and small-angle neutron scattering (SANS). Mentioned experimental techniques are suitable for detection of irradiation-induced microstructural changes at micro- and/or nanoscale level. However, one has to use a proper combination of more techniques (due to the complexity of the problem and weaknesses of the individual techniques) to obtain a complete picture.

Potential of PAS techniques can be seen in the evaluation of RPV steels microstructure. New information could be obtained via these non-destructive techniques application in the evaluation of defects concentration and its mobility due to different types of treatments.

Conventional PAS set-ups were used at our Institute of nuclear and physical engineering at the Slovak University of Technology for more than 25 years at different irradiated materials foreseen

for applications mostly in nuclear power industry. From several publications we would like to mention [4–7]. The radiation-induced precipitation effect of impurities in reactor pressure vessel (RPV) steels was dominant in our previous studies. Having in mind results from existing data of Slovak surveillance specimens programs collecting in collaboration with the utility operating 4 nuclear units (Slovenske elektrarne, plc.), we have investigated preferably the VVER-440 type reactor steels [1]. The main motivation of these studies was the optimization of the annealing procedure at the NPP. The annealing of 2 units in NPP Jaslovske Bohunice was performed in 1992 and the recovery of selected material properties was confirmed via Charpy-V or tensile tests. In addition, all surveillance specimens were characterized during 1–5 years before and after treatment, by transmission electron microscopy, Moessbauer spectroscopy and positron annihilation spectroscopy [1].

The knowledge and experiences obtained by extensive studies of RPV neutron embrittlement world-wide contributed to the development of new generations of RPV steels with improved chemical composition, where Cu a P are kept under strict limits 0.08 and 0.008 wt. %, respectively. Very inconvenient handling with neutron irradiated specimens, connected to the hot-cells cutting and polishing, turned the approach of our research to accelerating of irradiation via the experimental simulation by proper utilization of the light ion (hydrogen and helium) implantations [8–11].

It is well-known that RPV steels exposed to neutron irradiation in a reactor during the long-term operation will age over time via accumulated radiation damage [12]. The traditional method of studying radiation degradation effects is based on accelerated irradiation of model steels in research reactors. Nevertheless, it is necessary to mention, that not all results obtained from these condition can be implemented on the power reactors conditions. Firstly, the irradiation parameters (irradiation temperature, dose, flux) of a test reactor are generally difficult to control. Secondly, experiments within the power reactor neutron irradiation real conditions are usually extremely time-consuming (several years, special permissions by nuclear regulatory authority), expensive, and from the radiation protection point of view complicated (surveillance specimens are highly radioactive). It seems that a promising solution to these problems could be the use of light ions implantation as a surrogate for the neutron irradiation [13–17]. Relevant parameters of ions implantation can be much better controlled than the real neutron irradiation in the power reactor and the evolution of the damaged features can even be observed in situ. One of crucial factor is the penetration depth of ions implantation which depends on the type and energy of the particles used: electrons [18], light ions (H, He) [8,19,20], and heavy (or self) ions [21], the typical low energy proton beam or several MeV self-ions damaged layers have a thickness of the order of one micrometer below the surface of the irradiated sample. So, the detailed microstructural evolutions in this narrow modified layer (~μm) should be characterized by depth-sensing methods. PAS techniques are proven to be one of the most powerful and well-established tools for detailed characterization of vacancy-type defects in metals and alloys [22–25]. PAS using variable mono-energy positron beams allows one to probe the depth profile of a surface and near-surface defects in ion-irradiated structural materials [26], and has been successfully applied to study microstructural evolutions in RPV steels [27–31]. In the past 20 years, a growing body of evidence showed that the hydrogen ions implantation can emulate the neutron irradiation-induced microstructural damages, which provides a basis for its use in screening of new structural materials in current and future advanced reactors, while much fewer experiments have been done to benchmark Fe-ion implantation, which are technically more difficult but also worth for consideration at the higher energy treatment. In this review, we will use our unique results from differently treated German RPV steels (neutron irradiation as well as hydrogen ions implantation) from positron annihilation point of view. We would like to mention that we had the possibility to apply pulsed low energy positron beam, which is very suitable for the deep scanning of a relatively narrow region (for near surface damage study) where the implantation was performed. These newest results were correlated to our previous positron annihilation lifetime spectroscopy (PALS) studies on identical specimens treated by neutrons published in [8,32,33].

## 2. Experimental

For experimental studies, the originally SIEMENS (KWU) Germany specimens from the German research programs CARISMA and CARINA [34] were used. These RPV steel specimens were irradiated in German test reactor Kahl in the 70's. For our studies they were very suitable due to more than 35 years decay of [60]Co after irradiation. The irradiation temperature ranged in the scope 280–290 °C. The chemical composition of specimens is listed in Table 1.

**Table 1.** Chemical composition of CARINA/CARISMA studied steels in wt. % (Fe balance) [33,34].

| Specimen | German PWR Generation | C [%] | Si [%] | Mn [%] | P [%] | S [%] | Cr [%] | Mo [%] | Ni [%] | Cu [%] |
|---|---|---|---|---|---|---|---|---|---|---|
| P370 WM | 1 | 0.08 | 0.15 | 1.14 | 0.015 | 0.013 | 0.74 | 0.60 | 1.11 | 0.22 |

Two different specimens from the bulk material were selected and adapted for PAS measurements (from each material 2 pieces assembled in sandwich set-up). The specimens were irradiated to a neutron fluence of up to $2.23 \times 10^{19}$ cm$^{-2}$ [33]. We would like to note that the relatively high content of Cu and Ni in this steel negatively affects the final radiation damage.

Positron annihilation lifetime spectroscopy (PALS) is a well-established non-destructive spectroscopic method for evaluation of defect size (size of clusters) in materials and their density described by positron annihilation intensity. The sensitivity of PALS is relatively high with the ability to recognize one defect per $10^7$ atoms [35]. We were focused on the comparison of neutron irradiation and hydrogen ion implantation. The hydrogen nuclei were chosen for ion implantation due to almost the same mass of protons and neutrons and minimize the effect of injected interstitials, known from the heavy-ion bombardment experiments [36]. In order to minimize the dpa rate effect [20], the neutron damage was compared to hydrogen (proton) implantation of the same fluence. The irradiation conditions were, however, still very distinct, resulting in rather different defects in the microstructure. For our study, the implantation energy of hydrogen atoms was chosen on the level 100 keV. Based on literature search [37], we apply following parameters for the adequate SRIM (stopping and range of ions in matter) calculations: Displacement energy = 40 eV; Lattice binding energy = 0 eV; type of TRIM (transport of ions in matter) calculation—"Quick damage calculation".

Based on the SRIM calculations the maximum depth of damage in studied RPV steel specimens was about 0.64 μm and the maximum level of performed damage was in the depth of about 0.44 μm. The hydrogen ion implantation caused damage on the level of about 5 vacancies per ion. Three levels of implantation were performed. The third level of implantation was equal to the damage, from number of particles point of view, which was caused by neutron irradiation (see Table 1). For the sake of simplicity, dpa was averaged over the first 1 μm thick region. We would like to mention that we considered the fact that the ion implantation was performed only in relatively small depth range up to 640 nm and the neutron damage is spread throughout the whole volume of the specimens (see Table 2).

**Table 2.** Overview of implanted parameters in specimen.

| Hydrogen Implantation | Implanted Dose [C/cm$^2$] | Number of Implanted Ions [cm$^{-2}$] | Dose in Implanted Region [dpa] |
|---|---|---|---|
| 1. Level | 0.10 | $6.24 \times 10^{17}$ | 0.5 |
| 2. Level | 0.82 | $5.12 \times 10^{18}$ | 4 |
| 3. Level | 3.20 | $2.21 \times 10^{19}$ | 15 |

It is important to note, that the depth profile of positrons from the [22]Na source results in probing of a much deeper region than the one modified by 100 keV H implantation. In fact, only less than 10% of the positrons stop in the implanted layer, providing most of the information from the undisturbed bulk region [38]. Therefore, it is reasonable to reduce the obtained dpa values by one order of magnitude, giving the final "visible" damage of 0.05, 0.4 and 1.5 for the three levels of the implanted damage.

## 3. Results and Discussion

Previous PALS studies on neutron treated specimens [33] used two-component analysis characterized by parameters $\tau_1$ and $\tau_2$ and their intensities, respectively. The neutron irradiation leads to visible increase of positron mean lifetimes mostly due to higher positron trapping in the material. As the trapping places we could consider the Cu-rich solute clusters and/or vacancy-Cu complexes. The lifetimes $\tau_2$ (component assigned to the defect type characterization) of about 195–200 ps indicate the 1–2 vacancies presence. In the case of the steel P370 WM, the lifetimes in defects are higher and with values of about 203 and 213 ps. It could indicate vacancy clusters of 2–3 vacancies [33,39,40].

PALS results on hydrogen implanted specimens shown substantially higher level of damage. Differences to un-irradiated specimens were significant, but differences between neutron irradiated and hydrogen ions implanted samples (at the first 2 levels) were almost within the statistical error (Figures 1 and 2). Larger defects were observed at 3. level. Althouh specimens D77 and D161 were from P370 WM (characterized in Table 1), they were from different bulks prepared in different time. Differences could be partially explained probably via inhomogeneity due to slightly different preparing technologies.

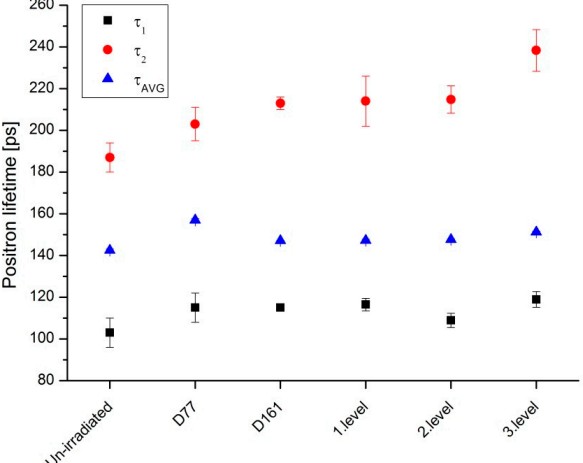

**Figure 1.** Positron lifetimes of all measured P370 WM specimens–unirradiated, neutron irradiated (D77 and D161) and 3 levels of H$^+$ ion irradiated.

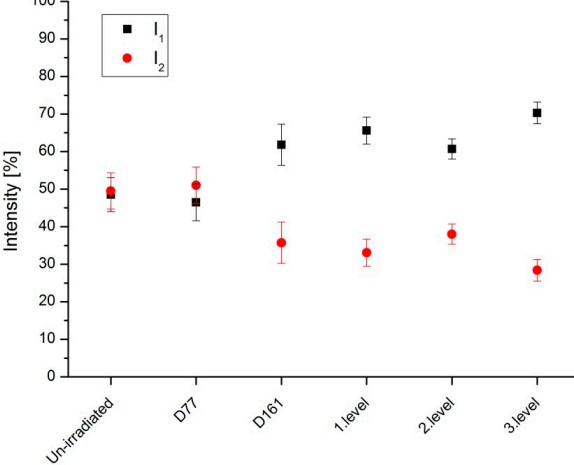

**Figure 2.** Intensity of positron lifetimes of all measured P370 WM specimens–unirradiated, neutron irradiated (D77 and D161) and 3 levels of H$^+$ ion irradiated.

Based on our previous studies performed on different RPV steels we would like to mention that for the ion implantation to the small depths up to 600 nm the conventional PALS technique is not very suitable and can detect only about 7–10% of the effect [41]. Therefore, for more precise verification, we measured described samples additionally also by Pulsed Low Energy Positron System (PLEPS) at FRM-II reactor in Garching neutron induced positron source Munich (NEPOMUC, Munich, Germany) [42–44]. Via this unique system we measured the depth profile of defects in about 20 steps in the studied surface region up to 600 nm. Currently, only several positron beam systems operated at an electron/positron injector linacs (KEK, Tsukuba, Japan) [45,46], (ELBE, Dresden-Rossendorf, Germany) [47] and PLEPS operated at the German research reactor FRM-II are dedicated to such surface experiment. Similar facilities could be perhaps used also at Kyoto University (Kyoto, Japan) [48] and McMaster University (Hamilton, ON, Canada) [43].

PLEPS results confirm our preliminary PALS results for *as-received* specimens. In the depth of about 500 nm, where back-diffusion to surface plays no role, the positron lifetime characteristics are typical for RPV steel, i.e., virtually all positrons annihilate as trapped at dislocations [49]. Big defects agglomerations, characterized by long lifetime components, are rare ($I_2 < 5\%$). After the implantation, the trapping at this type of defects, however, become the dominant annihilation route. In the implanted region of about 40 nm, the dramatic increase of the intensity of the second component ($I_2$) indicates a high concentration of large vacancy agglomerations. The lifetime over 400 ps at intensity of about 70% corresponds to an extremely damaged region. Despite the same fluence (about $2 \times 10^{19}$ cm$^{-2}$), the $H^+$ ion implanted sample was found to be extremely damaged, in comparison to the sample exposed to neutron irradiation. This not surprising considering the stopping profile of $H^+$ ions and the very narrow damage region—about 400 nm. Contrary to ionization losses, most of the of the energy was transferred into cascade collisions in near the Bragg peak. This resulted in a peak of primary and secondary knock-on atoms in the given depth, in contrast to the neutron irradiation which induced a uniform damage in the whole volume. The results of two-component analysis of the PLEPS lifetime spectra for as-received and $H^+$ ion irradiated samples are present in the following Figures 3 and 4:

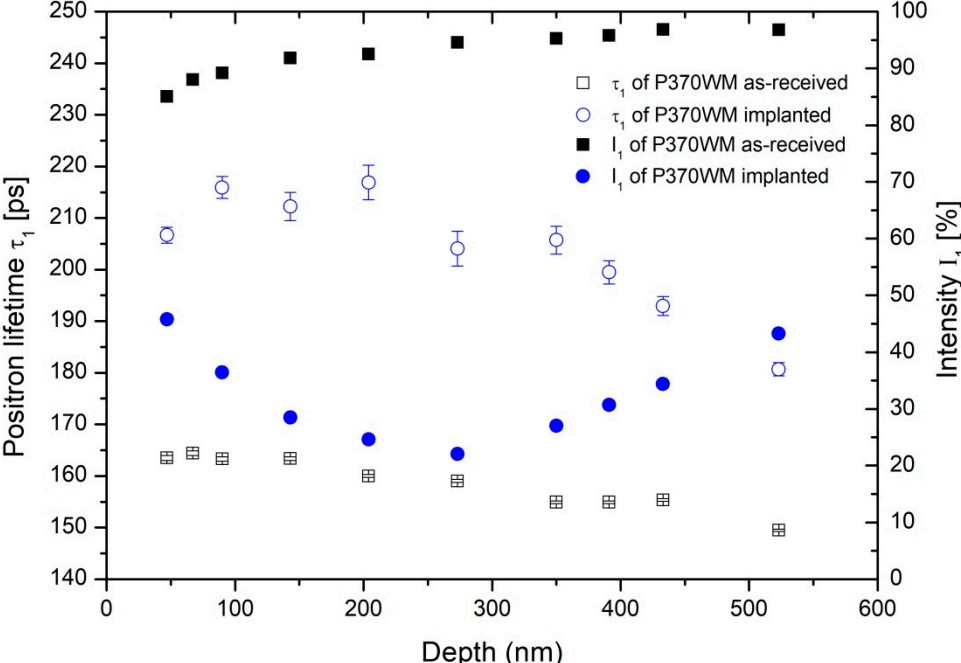

**Figure 3.** Comparison of PLEPS parameters in two components analyze of specimen P370 WM, before and after $H^+$ implantation by 100 keV (maximal damage at about 12 keV corresponding to depth of about 400–450 nm). Lifetimes and intensities of first component are presented.

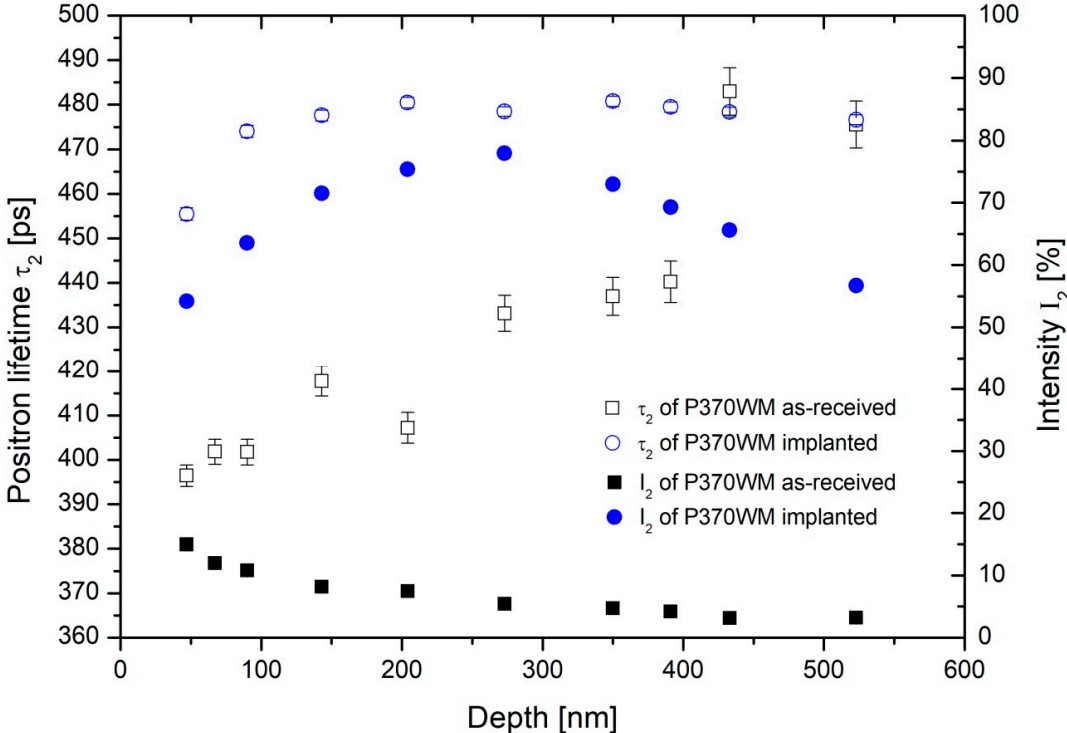

**Figure 4.** Comparison of PLEPS parameters in two components analyze of specimen P370 WM, before and after H$^+$ implantation by 100 keV (maximal damage at about 12 keV corresponding to depth of about 400–450 nm). Lifetimes and intensities of second component are presented.

Based on this analysis, it is possible to register a small increase in the first component lifetime. This can be explained by introduction of vacancies and small vacancy agglomerations, which accompanies the initial (as-received) dislocation-type defects, characterized by this complement in the pristine material. In as-received stage, the presence (intensity) of the first component was over 95% (see Figure 3) and practically characterizes the RPV bulk material. Comparing to the unirradiated condition, the intensity of the first complement decreased, due to a significant increase of the second complement. The second components could be assigned to large defects (agglomeration of about 50–100 vacancies) and in the as-received samples, its intensity was on the level of few percent. After the implantation, the intensity increased to 75% in the most damaged region (see Figure 4) and the positron trapping at large defect agglomerations dominated in the annihilation process. Considering the enormous increase of the mean positron lifetime by more than 200 ps near the damage peak depth (Figure 5), we can conclude that implantation of hydrogen ions on the level of 15 dpa practically destroyed the studied microstructure in the given region. In Figure 5 we show the increase of the mean positron lifetimes after hydrogen ion implantation as a function of the depth and compare it to dpa and hydrogen concentration profiles calculated by SRIM code.

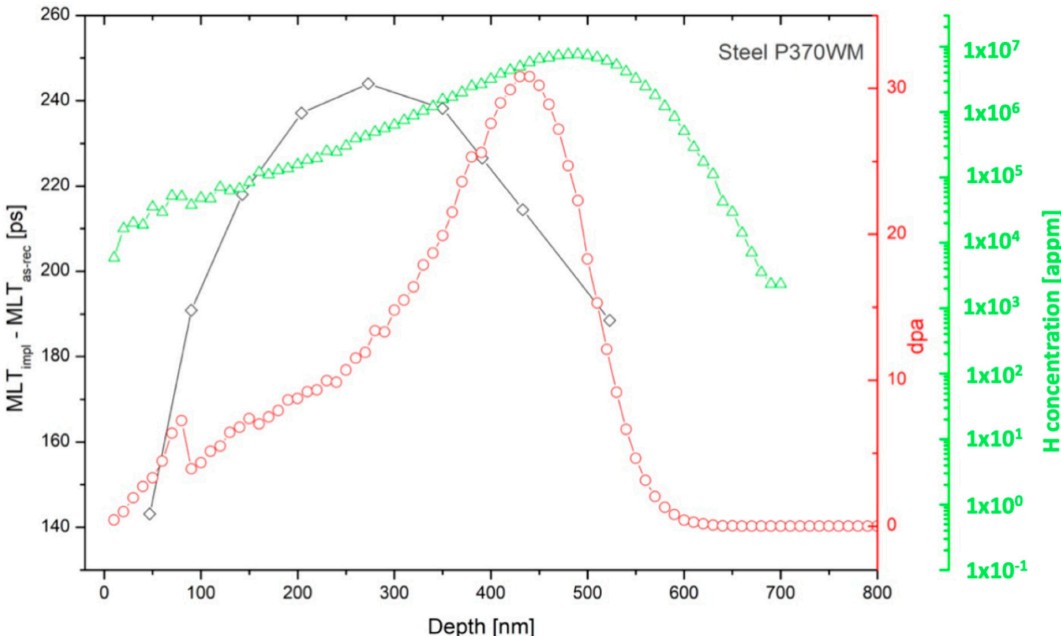

**Figure 5.** Comparison difference of positron mean lifetimes values before and after $H^+$ implantation of specimen P370 WM to dpa and H concentration profiles derived from SRIM code.

Results presented at Figure 5 shows a significant shift between the SRIM dpa peak and the observed positron lifetime changes. This can be interpreted by the hydrogen peak in the depth of ~500 nm which leads to a reasonable assumption of higher H-to-vacancy ratio in this region which effectively reduces the positron trapping at vacancy-type defects. In other words, the peak of positron trapping at these defects shall be expected in a shallower depth. This was indeed observed as shown in the Figure 5. The increase of positron lifetime along the decrease of intensity of the second component suggests that vacancy-hydrogen agglomerations grow via coalescence, resulting in larger cavities (high $\tau_2$) with lower number density (low $I_2$). A similar observation was reported by Dai et al. for 10–20 dpa Fe/Cr steels irradiated in spallation neutron target [50,51].

Although the dpa and H ion concentration have peak values between 450 and 500 nm (Figure 6), the H-to-dpa ratio increases exponentially with depth. Similarly, the intensity of the second defect component increases almost monotonically throughout the implanted region. This suggests that hydrogen is required for the stabilization of the radiation-induced defect clusters and with the increased ratio of hydrogen in vacancies, more of the clusters survive recombination processes in the irradiated microstructure.

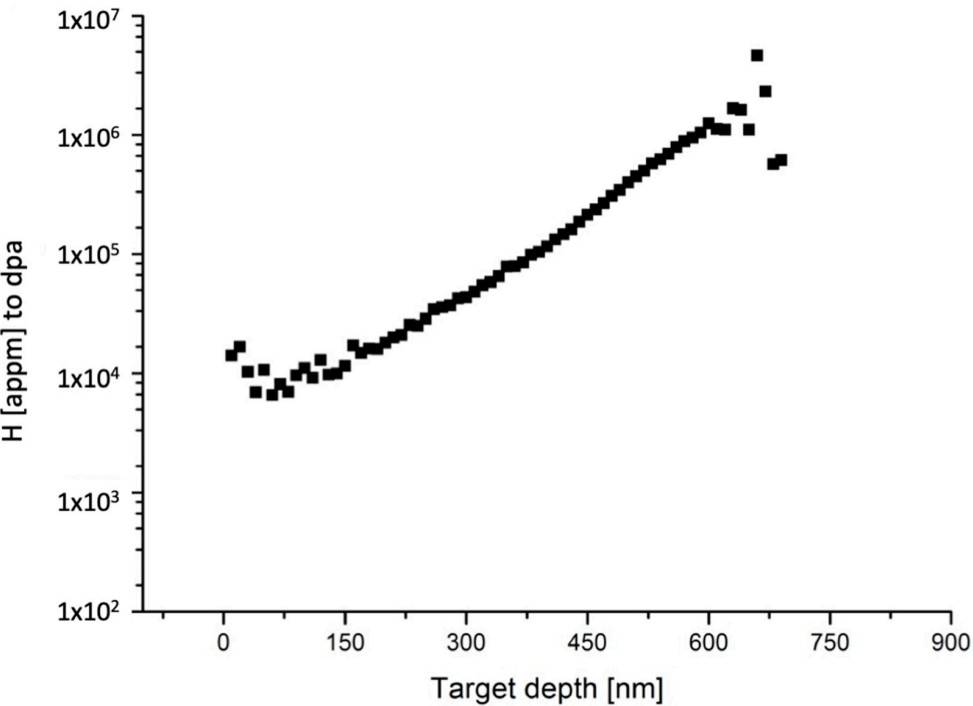

**Figure 6.** H/dpa ratio as the function of depth.

## 4. Discussion and Conclusions

Numerous studies in the past demonstrated that the reactor pressure vessel steel is sensitive to the alloying elements precipitation and to fast neutron flux. In theoretical explanation different authors reported [12,16,18] that the presence of copper atoms can cause under high neutron flux irradiation small clusters creation (d ~ 1–2 nm) which are surrounded mostly by P, Si or Mn atoms. It was confirmed that the phosphorus atoms segregate dominantly on Cu–clusters and dislocations and form P–clusters and/or atmospheres. Such mechanism leads to the RPV-steel embrittlement.

Due to proper thermal treatment (wet annealing procedure was performed successfully on several RPV including first two units in NPP Jaslovske Bohunice in Slovakia), a part of irradiation-induced copper and phosphorus clusters can be dissolved. Moreover, dissolution of Cu clusters occurs simultaneously with the growth of the Cu precipitates during NPP operation, too. The steel tensile properties, as well as the transition temperature, are substantially recovered due to thermal treatment.

In the present experiments, non-destructive PAS technique was in a complementary evaluation of the RPV-steels microstructure exposed to radiation environments. Via microstructural characterization of radiation-induced vacancy-type defects, which cannot be resolved by conventional TEM, we increase the general knowledge, which supports the scientific approach to the evaluation of the long-term operation of NPP units. Together with the previous studies of the radiation-induced Cu-P clusters or Cu-rich precipitation process, respectively, we outlined a complex insight into the behavior of the neutron-irradiated RPV steels. Small radiation-induced vacancy clusters were observed after all neutron doses in all studied specimens. These agglomerations of small point defects, visible only to positron annihilation techniques, cannot be excluded from the interpretation of the radiation effects on the physical and mechanical properties of the RPV steels. Nevertheless, they do not dramatically affect RPV lifetime (if they are distributed homogenously) due to external stresses like irradiation.

In this study, we have also performed an experimental simulation of neutron irradiation using the proton (hydrogen ion) irradiation. The primary aim of these experiments was to test the feasibility of such an approach and to avoid difficulties due to complicated radiation protection and limited handling with high activated samples. Hydrogen ion implantation was performed in 3 steps to the P370 WM samples which were representing the third generation of the RPV weld metal. While in the

neutron-irradiated samples short defect component of ~200 ps characterized the dominant annihilation sites (small vacancy clusters), in ion-implanted samples such component was found to have a relatively small intensity, comparing to a dominant long-lifetime component (>400 ps). Our results suggest that the long-lifetime component, attributed to clusters sized >1 nm, resulted from the coalescence of small vacancy clusters, described by the short defect components. It cannot be excluded that similar behavior would be observed in the same materials exposed to comparable displacement damage (dpa) introduced by neutrons, however, the effect of hydrogen as a vacancy-stabilizing gas must be considered. The not very well understood effect of implantation-introduced hydrogen and helium on the stabilization of radiation-induced vacancy-type defects is still the major drawback of the using of ions as surrogates for neutron irradiations.

From the nuclear plant lifetime management point of view, it seems to be positive that no large voids or vacancy clusters were observed due to neutron irradiation treatment for the given fluencies. This indicates that there are not signals for the necessity to limit (not to extend) the operational lifetime of German NPPs from the RPV point of view.

**Author Contributions:** Conceptualization, V.S.; data curation, W.E., S.S. and J.S.V.; formal analysis, V.K.; funding acquisition, V.S.; investigation, V.S., V.K..; methodology, V.S. and V.K.; resources, M.P. and J.S.V.; supervision, V.S.; writing (original draft), V.S. and V.K.; writing (review and editing), V.S. All authors have read and agreed to the published version of the manuscript.

**Funding:** This study was granted by VEGA 1/0104/17 and by Slovak Research Agency (project 313011U413 in frame of OPVaI-VA/DP/2018/1.1.2-01).

**Acknowledgments:** The authors would like to thank to the Heinz Maier-Leibnitz Zentrum (MLZ), Garching, Germany.

**Conflicts of Interest:** The authors declare no conflict of interest.

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
