# Peer review of "Radiation Damage of Reactor Pressure Vessel Steels Studied by Positron Annihilation Spectroscopy—A Review"

_metals, doi:10.3390/met10101378_

Round 1

Reviewer 1 Report

The paper describes the possibility to use no-destructive method to evaluate the defects in irradiated steel components from nuclear reactor vessels. The possibility to replace long neutron irradiation by a much quicker and more convenient (less radioactivity) ion implantation is also considered.

My main two suggestions to the authors are:

1 Please explain in the introduction how the PAS can be used to characterize the microstructure (measurement of two decay times?) and what characteristics are determining these time constants. Since all the results are based on PAS measurements I believe that is important for this paper as many readers of the article will be not familiar with that specific technique.

2.After reading the article it is not clear to me whether the ion implantation can be used as an evaluation tool for the characterization of longevity and stability of materials for the future reactors. Just stating that this question was considered is not a good statement for the paper, please add some conclusions in the abstract.

-----------------------------------------------------------------------------------

Abstract: “This paper is a review of experimental PALS/PLEPS” – please spell PALS and PLEPS

Abstract: “The possibility of substitution of neutron treatment via hydrogen ions implantation was 20 analyzed as well.” – this reads as H ion implantation can help in making materials last longer, while this is only considered to substitute irradiation to speed up the metal ageing for the studies of component longevity only. Suggest to rephrase that sentence.

Line 40: change to “Based on [1,2] the most important degradation factors ARE:”

Line 46: is that a common term “irradiation temperature”? I am more used to irradiation spectrum or energy distribution of neutron radiation

Lines 57-70: “The basic 55 requirements for the RPV materials are the following:” – then the list does not really spell the requirements. Thermal properties – not a requirement, Technological properties – not a requirement, etc. Please rephrase that sentence as to state which properties of materials are important or something like that.

Line 130: “…while much fewer experiments have been done to benchmark Fe-ion implantation.” – please explain here why suddenly Fe-ion implantation is of interest and how it is related to the damage to the materials in power reactors.

Line 122-137: Not clear to me how that very thin layer (only ~ 1 um) emulated the damage to steel which is more evenly distributed over the volume of the steel component. Does very localized nature of simulated ageing represent what happens in the bulk in full reactor materials? Please add a comment here.

Line 143: “The irradiation temperature ranged in the area 280 - 290 °C.” – temperature range is not an area, correct please.

Line 165: “and the maximum level of performed damage was about 0.44 μm.” – the 0.44 is the depth where the damage was the highest.

In section 1.experimental I suggest to add few sentences on what is actually measured by PAS and how that gives the information on the defects, that will be very helpful for the understanding of how these results of your study were obtained.

Line 181: that should be section 2, not 1.

Line 183: two parameters are mentioned here and not explained at all what they mean and how they are measured, only reference is given. Please add a brief description on what they are.

Fig. 1. I do not understand why Tau_average for D77 is the largest while both Tau 1 and Tau2 for 3. Level are both higher than in case of D77? How Tau_average is calculated then?

Line 190-192: “Differences to un-irradiated specimens were significant, but differences between neutron 190 irradiated and hydrogen ions implanted samples were almost within the statistical error (Figs. 1 and 2).” – it is not what I see in these figures, disagree with these statements.

Only 3. Level shows substantially larger time constants, the rest of them are very similar to irradiated sample D161

No explanation on what are D77 and D161 samples. Need to have that. They show very different results, please explain why. How these specific samples were chosen?

Line 273: should be section 3.

Line 274: sensitive on – wrong English

Conclusions: “Small radiation-induced vacancy 292 clusters were observed after all neutron doses in all studied specimens. These agglomerations of 293 small point defects, visible only to positron annihilation techniques, cannot be excluded from the 294 interpretation of the radiation effects on the physical and mechanical properties of the RPV steels” – I do not understand how that is related to the longevity of steel component? It is stated that no large clusters were observed and that supports the extension of lifetime for the specific reactors. What about these small clusters which were seen everywhere? How do they affect the integrity of steel?

To a person who is not well familiar with PAS methods, I do not see from the measured results presented in the paper that there were no large clusters observed in all samples? Where is that shown in the data? Since that is one of the most important (largest implications) conclusions I suggest the authors explicitly write on how that conclusions was reached with data measured in this study.

Did I also understood it correctly that ion implantation does not represent the damage due to neutron irradiation? The measurements showed that for irradiated steels the  annihilation sites had short decay times of 200 ps (small vacancy clusters), while implanted materials have >400 ps  times corresponding to larger clusters.

In abstract it is stated that ion implantation was considered. I think the authors should state then the conclusion: not only mention that it was considered, but rather state that ion implantation cannot replace irradiation, which still needs to be done despite all the complications numerated by the authors in the introduction.

Author Response

Remarks are accepted and will be incorporated in the revised version.

Reviewer 2 Report

Comments:

The average positrons lifetimes and intensities were determined using the PALS technique. This made it possible to estimate the vacancy clusters size. However it is known that radiation-induced point defects make a contribution to the steel hardening, which depends both on its barriers strength factor to dislocation movement and on point defects density and size. In this connection it is recommended to estimate vacancy clusters density and average size to be able to assess their contribution to the yield stress change according Orowan equation. It is also possible to carry out a comparative analysis of vacancy clusters density and sizes changes depending on radiation dose. These parameters are of practical importance for assessing contribution of the radiation-induced secondary phases and radiation defects (dislocation loops) compared to contribution of the vacancy clusters to radiation embrittlement of reactor vessel steels.   

Author Response

Reviewer remarks were accepted and the text will be improved in next version.

Improvements via grammar correction or explanations to both reviewers remarks are indicated yellow.
